# Mattering and Depressive Symptoms in Portuguese Postpartum Women: The Indirect Effect of Loneliness

**DOI:** 10.3390/ijerph191811671

**Published:** 2022-09-16

**Authors:** Bárbara Caetano, Mariana Branquinho, Maria Cristina Canavarro, Ana Fonseca

**Affiliations:** 1Faculty of Psychology and Educational Sciences Coimbra, University of Coimbra, 3000-115 Coimbra, Portugal; 2Center for Research in Neuropsychology and Cognitive Behavioral Intervention, Faculty of Psychology and Educational Sciences, University of Coimbra, 3000-115 Coimbra, Portugal

**Keywords:** depressive symptoms, loneliness, mattering, postpartum, postpartum depression

## Abstract

Background: Postpartum depression is described as the most prevalent clinical condition in the postpartum period, with several negative consequences. The current study aimed to understand the relationship between mattering, loneliness and depressive symptoms in Portuguese postpartum women and to examine the potential mediating role of loneliness in the relationship between mattering and depressive symptomatology among postpartum women. Methods: This cross-sectional study included a sample collected online composed of 530 Portuguese women in the postpartum period, who answered self-report questionnaires to assess depressive symptoms, mattering, and loneliness. Results: It was found that the relationships between mattering, loneliness, and depressive symptoms were significant, *p* < 0.001: (a) higher levels of mattering were associated with lower levels of loneliness and depressive symptomatology and (b) higher levels of loneliness were associated with higher levels of depressive symptomatology. The relationship between mattering and postpartum depressive symptoms occurred directly and indirectly through loneliness, 95% CI = [−0.75, −0.46]. Conclusions: These results highlight the importance of studying loneliness as a possible risk factor for postpartum depression and alert to the pertinence of considering mattering and loneliness in the assessment and intervention with women in the perinatal period.

## 1. Introduction

The transition to motherhood is increasingly recognized as a demanding period associated with high-stress levels, whose changes and challenges can act as precipitating factors for the development (or recurrence) of mental health problems in women [1]. Postpartum depression [PPD] is the most prevalent clinical condition in the postpartum period, affecting about 10 to 15% of women [2]. There are several negative consequences associated with PPD, namely in the mother–child relationship, and in the child’s cognitive, emotional, behavioral, and social development [3], as well as in its attachment style [4]. Additionally, it may affect the relationship with the partner, who reports feeling oppressed, frustrated, angry, and helpless [5].

### 1.1. Loneliness and Depressive Symptoms in the Postpartum Period

Loneliness can be conceptualized as a painful and subjective experience of the absence of social contact, lack of belonging to a group or feelings of isolation [6]. Since March 2020, due to the COVID-19 restrictions, the process of socialization and face-to-face support for pregnant and postpartum women became scarce [7]. During and after childbirth, the presence of the baby’s father or another person in the delivery room was no longer possible and hospital visits were prohibited. In addition, home visits by family and friends were restricted, not only due to social distancing measures but also because postpartum women reported being afraid of putting the new-born at risk of contracting the COVID-19 virus [7,8]. According to the existing research, these restrictions have resulted in increased isolation and lack of support from family and friends, which has contributed to feelings of loneliness and depressive symptoms [7,8]. Several studies have found that the prevalence of depressive symptoms in postpartum women increased compared to the period before the pandemic [9,10]. According to Zanardo and colleagues, about 30% of women who had children during COVID-19 pandemic had depressive symptoms with clinical significance, compared to 11.9% of women who had children the year before the pandemic [10]. A study conducted in Canada with pregnant women and women in the first year after delivery found a higher prevalence of depressive symptoms in these women during pandemic (40.7%) in comparison to pre-pandemic period (15%) [11]. Similarly, according to a study carried out in Israel, there was an increase in the rates of depressive symptoms among women who gave birth during the COVID-19 pandemic [12]. Furthermore, a study conducted in Italy reported that postpartum women who remained isolated in the most affected area—in terms of contagions and deaths—experienced a greater increase in the level of depressive symptomatology compared to those in the least affected areas [13]. Moreover, postpartum women reported that the fact they could not share part of their experience with their family, their social support network, and other parents made them feel that part of their experience as mothers was being taken away from them, leading to feelings of isolation and loneliness [7].

Loneliness and social isolation may negatively affect physical and mental health [14]. It is considered that loneliness is one of the greatest risk factor for the occurrence of depression [15,16], and that higher levels of loneliness are associated with higher levels of depressive symptomatology [17,18,19]. It is also important to highlight that depression is likely to influence the way individuals perceive the available social support as insufficient and thus may increase the feelings of loneliness [18]. In this context, a recent study with pregnant women showed that loneliness mediated the relationship between perceived social support and depression [20]. Therefore, loneliness began to be pointed out as an important factor for understanding PPD [21]. According to a study conducted with Indian pregnant and postpartum women, the proportion of those who reported feelings of loneliness was higher in women who had PPD (26.3%) than in those who did not have a clinical diagnosis of PPD (6.9%) [22].

### 1.2. Mattering and Depressive Symptoms in the Postpartum Period: The Potential Mediating Role of Loneliness

Mattering concerns the tendency to consider ourselves significant to others, i.e., the belief that we are important to people who are significant to us. This construct has two dimensions, namely general mattering (importance we have in society or the community) and interpersonal mattering (importance we have for significant people, how they pay attention to our needs and show interest in them) [23].

Several studies indicated that lower levels of mattering were associated with higher levels of depressive symptoms [24,25]. According to a study conducted in general population, changes in levels of mattering were predictive of changes in depressive symptomatology and the relationship between these variables is conditioned by gender (i.e., changes in the mattering were associated with variations in depression in women but not in men) [25]. Effectively, the perception that one is important to others is associated with a decrease in depressive symptoms [26].

According to our knowledge, there are no studies about the relationship between mattering and PPD. However, there are changes in interpersonal relationships (with the partner, with the family of origin, with friends) after childbirth that may influence how women perceive the degree of importance they have for others. Specifically, support needs in the postpartum period are different compared to other periods of life due to the demands of caring for a new-born, and women often find that family and friends are less present than expected, evaluating their social interactions as more negative [27].

In addition, lack of mattering to others—i.e., feeling that others do not care about us and that we are not a source of interest on their part—can lead us to experience a sense of loneliness [28]. Recently, it has been hypothesized that the mattering is related to loneliness [28,29]. However, to our knowledge, there are only two studies evaluating the relationship between these two constructs. One study found a negative and robust association between mattering and loneliness, which corroborated the hypothesis that lower feelings of mattering are associated with higher levels of loneliness [29]. Congruently, it has also been found that mattering acted as a predictor of loneliness in a sample of young women aged between 18 and 25 years [28].

### 1.3. The Present Study

Considering that mattering and loneliness are associated with mental health problems, and a lower perception of mattering has been shown to predict higher levels of loneliness, it is important to understand how the relationship between mattering and loneliness can influence women’s mental health in the postpartum period. Specifically, this study aimed: (a) to understand the relationship between mattering, loneliness, and depressive symptomatology in Portuguese women in the postpartum period; and (b) to examine the potential mediating role of loneliness in the relationship between mattering and depressive symptomatology among postpartum women (Figure 1).

## 2. Materials and Methods

### 2.1. Procedures

This study was a part of a wider cross-sectional study entitled identification and characterization of different subtypes of postpartum depressive symptomatology and evaluation of the acceptability of blended interventions for postpartum depression that was approved by the Ethics Committee of Faculty of Psychology and Educational Sciences of University of Coimbra.

The sample collection occurred between June 2020 and February 2021. The inclusion criteria to participate in this study were: (a) being a woman in the postpartum period (i.e., having been a mother in the last 12 months), (b) being 18 years old or older, and (c) being Portuguese and living in Portugal. The dissemination of the study was conducted on social networks—Instagram, Facebook, and LinkedIn, and in groups and forums focused on parenting. In addition, informative leaflets containing information about the study and a QR Code to access the questionnaire were placed at health institutions (e.g., the Tondela-Viseu Hospital Center (maternity hospital) and the São Cosme Clinic, Viseu).

Participants were asked to complete a self-response questionnaire hosted in the LimeSurvey^®^ software. The objectives underlying the study, the role of participants and researchers, and the confidentiality and anonymity aspects of the study, were described on the first page of the questionnaire. Before accessing the evaluation protocol, the participants gave their informed consent to participate in the study, by indicating the option “I agree to participate in the study”. It was also given the information that they could give up at any time if they wished. Participation in the study was voluntary, and there was no compensation for participation.

### 2.2. Sample

The sample consisted of 530 postpartum women from the community, aged between 19 and 49 years old (mean age = 33, *SD* = 4.94). Most women were married/cohabiting (*n* = 481, 90.8%), lived in an urban area (*n* = 395, 74.5%), and were first-time mothers (*n* = 362, 68.3%). Concerning socioeconomic characteristics, most women had higher education (*n* = 376, 71%), were employed (*n* = 400, 75.5%), and had an average monthly income between 500 € and 1000 € (*n* = 249, 47.0%).

Regarding clinical characteristics, 31.3% of women (*n* = 166) reported having previous history of psychological or psychiatric problems and 36.8% (*n* = 195) had psychological or psychiatric treatment in the past.

The mean age of the babies was 5.22 months (*SD* = 3.38) and varied between 0 and 12 months. They were born between 31 and 42 weeks of gestation (*M* = 38.98, *SD* = 1.57). Of the 530 participants, 44 (8.6%) report that their baby has been detected with a chronic health problem.

### 2.3. Measures

#### 2.3.1. Sociodemographic, Clinical, and Infant’s Characteristics

Sociodemographic (e.g., age, marital status, number of children, academic qualifications, employment status, monthly income, residence) and clinical information (e.g., history of psychopathology), as well as infant’s characteristics (e.g., baby’s age, gestation week at birth, or if some medical problem has been detected in the baby), were assessed through a self-report questionnaire developed by the research team.

#### 2.3.2. Edinburgh Postnatal Depression Scale (EPDS)

The EPDS [30,31] was used to assess the presence and severity of depressive symptomatology in the last seven days. EPDS is a self-report questionnaire composed of 10 items (e.g., “I had ideas of doing harm to myself”), answered on a 4-point response scale ranging from 0 to 3, according to the increasing severity of the symptoms. Total score can vary between 0 and 30 points. According to the Portuguese validation of EPDS [31], a total score higher than nine points is indicative of the presence of clinically significant symptomatology. In our study, the internal consistency value of the scale was high (Cronbach’s α = 0.90).

#### 2.3.3. Loneliness Scale (ULS-6)

The ULS-6 [32] aims to evaluate the subjective feeling of loneliness. It is a one-dimensional scale, composed of six items (e.g., “I feel like I’m part of a group of friends”), answered on a 4-point Likert scale ranging from 1 (*Never*) to 4 (*Often*), and the total score can vary between 6 and 24. Higher scores are indicative of a higher subjective feeling of loneliness. The USL-6 has good psychometric properties, emphasizing its high internal consistency (Cronbach’s α = 0.82). In our study, the scale’s internal consistency was high (Cronbach’s α = 0.88).

#### 2.3.4. General Mattering Scale (GMS; Psychometric Studies of Portuguese Version Ongoing)

The GMS [33] is a one-dimensional scale that assesses the subjective degree to which the participant believes to be important to others. It consists of five items (“How do you feel that people depend on you?”) answered on a 4-point response scale ranging between 1(*Nothing*) to 4 (*A lot*). The total score varies between 5 and 20, and higher scores indicate a greater perception that the subject is important to others. In our study, the internal consistency was high (Cronbach’s α = 0.87).

### 2.4. Data Analysis

Statistical analyses were performed with the software Statistical Package for the Social Sciences (IBM SPSS, version 25.0, Armonk, NY, USA). Descriptive statistics were calculated to characterize the sample concerning its sociodemographic, clinical, and infant-related characteristics. Pearson’s correlation coefficients were computed to examine the associations between variables (mattering, loneliness, and depressive symptoms). Spearman’s correlation coefficients were calculated to evaluate the relationships between sociodemographic, clinical, and infant variables (age, education level, residence, baby’s age, number of children, and gestation week at birth), and depressive symptoms, in order to identify the variables to be introduced as covariables in the models analyzed. The magnitude of the effect of these correlations can be classified as low (*r* < 0.30), moderate (0.30 < *r* < 0.50), or high (*r* > 0.50) [34]. Due to the existence of missings in covariables—particularly in education level—the sample was reduced to 515 women. A simple mediation model was tested to assess the direct and indirect effects on the relationship between mattering and depressive symptoms, using the SPSS Macro PROCESS [35]. Mattering was introduced as an independent variable, the depressive symptoms as a dependent variable, and the loneliness as a mediating variable. Sociodemographic, clinical, and infant’s characteristics were introduced as covariates if they were significantly associated with depressive symptoms (*p* < 0.05).

The indirect effect was estimated through the bootstrapping procedure (with 5000 resamplings), estimating confidence intervals of 95% (CI; Bias-Corrected and Accelerated Confidence Intervals 95% IC). The indirect effect was considered significant if zero value was not within the CIs range.

## 3. Results

### 3.1. Association between Mattering, Loneliness, and Depressive Symptomatology

As presented in Table 1, it was found that mattering was significantly and negatively associated with loneliness and with depressive symptomatology (moderate and high correlation, respectively), i.e., higher levels of mattering were associated with lower levels of loneliness and depressive symptomatology. In addition, higher levels of loneliness were associated with higher levels of depressive symptomatology.

### 3.2. Potential Mediating Role of Loneliness in the Relationship between Mattering and Depressive Symptomatology in Portuguese Women in the Postpartum Period

#### Preliminary Analyses

Correlations were made between some sociodemographic, clinical, and infant characteristics (age, educational level, residence, baby’s age, number of children, and gestation week at birth) and depressive symptomatology. Overall, it was found that age (*r*_s_ = −0.10, *p* = 0.024), educational level (*r*_s_ = −0.17, *p* = 0.001), and number of children (*r*_s_ = −0.09, *p* = 0.043) were significantly correlated with depressive symptomatology in the postpartum period (see Table 2). Thus, they were introduced as covariables in the mediation model.

### 3.3. Direct and Indirect Effects of the Relationship between Mattering and Depressive Symptomatology in the Postpartum Period

Figure 2 presents the model referring to the direct and indirect effects of the mattering on depressive symptoms in the postpartum period. In this model, mattering and loneliness explained 34% of the variance of depressive symptomatology.

For simplicity, measurement error terms, nonsignificant paths, and the correlation of covariates with the dependent variables are not presented. The estimate of the correlations between education level (*B* = −0.81, *p* = 0.001) and number of children (*B* = −0.63, *p* = 0.044) with depressive symptoms were significant.

Mattering significantly influenced loneliness, explaining 47% of the variance of this variable. In addition, loneliness and mattering were significantly associated with depressive symptoms. The associations between the covariables introduced in the model (age, educational level, and number of children) and loneliness were not significant. However, education level (*B* = −0.81, *SE* = 0.24, *t* = −3.33, *p* = 0.001) and number of children (*B* = −0.63, *SE* = 0.31, *t* = −2.02, *p* = 0.044) were negative and significantly associated with depressive symptoms.

Total and direct effects of mattering on depressive symptoms were statistically significant. Therefore, lower levels of mattering were associated with higher depressive symptoms in the postpartum period.

It was also found an indirect effect of mattering in depressive symptomatology through loneliness, *B* = 0.60, *SE* = 0.07, 95% CI = [−0.75, −0.46], which means that the relationship between mattering and depressive symptoms in the postpartum period occurred directly, but also indirectly, through the experience of higher levels of loneliness. In other words, lesser mattering contributed to a greater subjective experience of loneliness, which led to higher levels of depressive symptomatology.

## 4. Discussion

This study presents innovative findings, contributing to a better understanding of the relationship between mattering, loneliness, and depressive symptomatology in Portuguese women in the postpartum period. The results of this study revealed that lower levels of mattering negatively influenced depressive symptomatology in Portuguese women in the postpartum period, either directly or indirectly through the perception of higher levels of loneliness.

Our results also revealed that the perception that one is important to others is associated with depressive symptoms. Similarly, a study conducted with first year University students found that negative appraisals of mattering were associated with elevated levels of depressive symptoms [32]. This also seems to be true in the postpartum period, since the birth of a child is associated to changes in interpersonal relationships (with the partner, with the family of origin, with friends) that may influence how women perceive the degree of importance they have for others. Specifically, support needs in the postpartum period are different compared to previous periods of life due to the demands of caring for a new-born. Since the study was carried out during the peak of COVID-19 pandemic, it is considered that the social restrictions inherent to COVID-19 [7,8] may have contributed to women feeling that their support needs were not being met, leading to a lower perception of mattering. Effectively, not feeling important to others contributes to the formation of a negative view of the self, which leads a person to evaluate himself as someone who is not pleasant to live with, which can be translated into an increase in depressive symptomatology [36].

Our results also revealed an indirect effect of loneliness on the relationship between mattering and depressive symptomatology. In fact, a lower perceived importance to others can lead to the development of maladaptive strategies in relating to others, which contributes to greater social detachment and to an experience of loneliness and isolation [36]. Those who experience higher levels of loneliness are more likely to have a more negative view of self, others, and the world and expect social interactions to be unfruitful [37], and this is associated with the experience of depressive symptomatology.

Although there are few studies about the effect of loneliness on depressive symptomatology in the postpartum period [21,22], a possible explanation for the relationship between loneliness and PPD may be that women feel disconnected from their habitual routine, missing their previous way of life, which may contribute to the development of depressive symptoms in the postpartum period [21]. During this period, women often tend to compare themselves or be compared with other mothers, and often receive unsolicited critics and opinions from their families and other mothers, which may contribute to them feeling judged, ashamed, vulnerable, and guilty about not meeting the socially imposed expectations of “perfect motherhood” [27,38]. This can contribute for women to feel alone and creating a barrier in relationships with others, reinforcing their levels of loneliness [27]. According to a qualitative study [38], conducted with women that were mothers of children up to three years and that had experienced postpartum anxiety, most of them felt that their partners’ support was not enough, and that the division of tasks was not equitable. This overload of tasks in the care of a newborn, often accompanied by caring for other children, can lead to feelings of loneliness. Several studies postulate that a subjective feeling of loneliness triggers in the individual who experiences more negative evaluations of emotions and low satisfaction with life [39,40], which can lead to an increase in depressive symptomatology. Feelings of loneliness may have been increased by the context of COVID-19 pandemic. The fact that confinement did not allow contact with close family and friends, the limited possibility of access to preparatory and postpartum courses, the lack of contact with other mothers and sharing of experiences led to a decrease in the perception of social support. This may have increased the feeling of loneliness among women in the postpartum period and the prevalence of PPD [7,8].

It was also found that education level and the number of children was associated with depressive symptomatology, i.e., a greater education level and higher number of children were associated to lower levels of depressive symptomatology experienced by women during the postpartum period. Regarding education level, the results of our study are congruent with the results of a study conducted by O’Hara and McCabe (2013), who found that women with fewer educational qualifications present a higher risk of developing PPD [2]. One possible explanation for this association is that women with a greater education level usually have a better monthly income, which gives them a better socioeconomic status. In addition, being more literate can help them have better access to prepartum and postpartum courses, which are essential for both pregnancy and postpartum planning.

Although in our study there is a significant association between the number of children and depressive symptoms, this association was not found in other studies [41]. This result may be due to the fact that most of our sample consisted of women who were first-time mothers. Alongside this, it is hypothesized that COVID-19 also contributed to these results. It is hypothesized that women who were mothers for the first time may have experienced higher levels of depressive symptomatology, since the transition to motherhood is characterized as a period of crisis, that with the assumption of a new role in the woman’s life and all the physical and psychological changes that it entails, concomitantly with the reorganizations in the family and personal sphere, can originate a feeling of loss of identity and lead to the questioning of the meaning of life [42]. Specifically, primiparous women must reorganize themselves to adapt and answer a baby’s needs (e.g., food, hygiene, and sleep). In addition, these women were confronted with the need to face another crisis, the pandemic period, which with all its specificities and consequences, can lead to more significant adaptation difficulties and, consequently, higher levels of depressive symptomatology. However, this is a topic that still needs research.

### 4.1. Limitations and Future Research

Despite the relevant findings, this study also presents some limitations that should be mentioned and considered in the interpretation of the results. First, although the directionality of the relationships proposed in the mediation model under study was based on empirical research [21,28,29,43], causality relations between the variables cannot be assumed due to the cross-sectional nature of the study. Additionally, they were assessed through self-response questionnaires, which does not allow variables to be directly observable. It would be important for future research to replicate these results through a longitudinal study to determine the meaning of the relationship between the study variables and to help understand how they are established during the first year after childbirth.

Second, the sample was collected online (through dissemination on social networks and online forums) and was self-selected (i.e., women who had more interest in this topic may have filled out the questionnaire). In addition, the sample was mainly composed of highly educated women (71%), married/cohabiting women, and women resident in urban areas. Therefore, the sample may not be fully representative of the entire Portuguese population in the perinatal period.

Third, although it is determined by clinical consensus that depressive symptoms in the postpartum period can occur up to 12 months after delivery [2,44], the specific circumstances of the postpartum period may vary over time. Considering the study’s cross-sectional nature, it was not possible to capture the variability of the challenges experienced by women across the postpartum period, and how this could be translated into changes in the variables under study over time. The conduct of longitudinal studies will be fundamental to clarify this topic.

Although in our study we did not assess specific variables related to COVID-19 (e.g., perceived risk, the degree to which postpartum women were affected by the protective measures implemented), it acted as an important contextual factor. In fact, we hypothesize that the resulting restrictions and conditions could have influenced our results and therefore future research conducted at a post-pandemic time should replicate this study, to understand the magnitude of this influence.

### 4.2. Implications for Practice

Regardless of the limitations pointed out, knowledge about the relationships between mattering, loneliness, and depressive symptomatology contributes to clinical practice during postpartum period. On the one hand, it was demonstrated the relevance of studying loneliness as a possible risk mechanism for PPD, considering its influence on women’s depressive symptoms in the postpartum period and in the relationship between mattering and depressive symptomatology. These results also demonstrated the relevance of considering these two constructs (mattering and loneliness) during the evaluation and intervention of women with PPD, and the importance of implementing intervention programs and groups support for pregnant and postpartum women, to prevent or reduce loneliness and depressive symptoms.

Our results highlight the importance of interpersonal relationships for women during pregnancy and postpartum period. Given that mattering and loneliness can influence depressive symptomatology in the postpartum period, it would be important to implement the evaluation of these two constructs during pregnancy, maximizing the possibility of adopting prevention measures (e.g., preparation and psychoeducation of the primary sources of support yet in pregnancy, in which the medical community and healthcare provider are integrated; promotion of cognitive flexibility strategies to generate more realistic and functional interpretations of the perception of importance to others and support received, since in some cases it is the interpretation of the woman which is biased, due to the presence of cognitive distortions). Overall, by identifying mattering and loneliness as potential risk factors for PPD, our study brings important implications for practice, since health professionals can be more rigorous during screening, as well as taking preventive measures for this clinical condition [2,45].

## 5. Conclusions

Our study added to prior research by showing significant associations between mattering, loneliness, and depressive symptoms in a sample of Portuguese women in the postpartum period. This study demonstrated that mattering was significantly associated with depressive symptoms, directly and indirectly, through loneliness, i.e., a lower sense of being important to others leads to a higher subjective experience of loneliness, which lead to higher levels of depressive symptomatology. Our results highlight the importance of assessing loneliness as a possible risk factor for postpartum depression and the relevance of mattering and loneliness in the intervention with postpartum women (e.g., support groups, education campaigns to general population with particular focus in women support network, intervention programs, implementation of cognitive flexibility strategies). In addition, our study highlights the great importance of interpersonal relationships and social support for women during the postpartum period.

## Figures and Tables

**Figure 1 ijerph-19-11671-f001:**
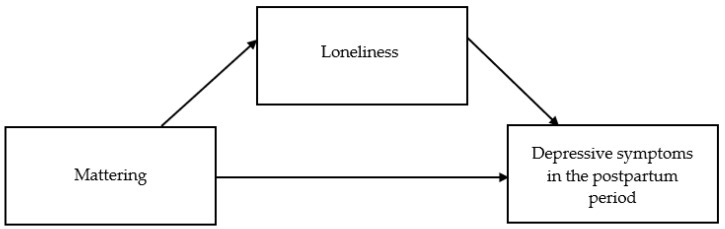
Conceptual scheme of the study.

**Figure 2 ijerph-19-11671-f002:**
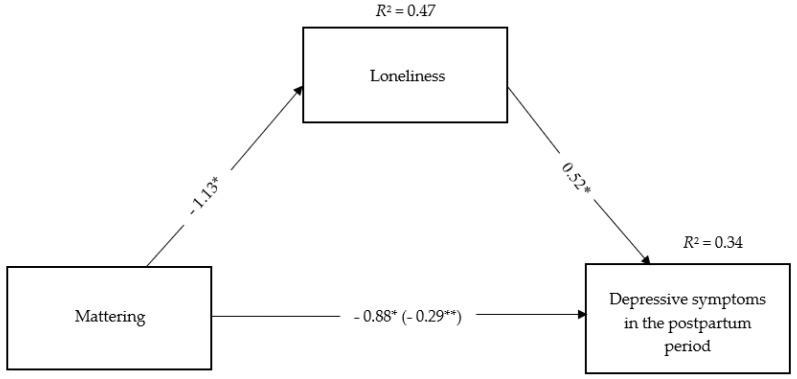
Direct and indirect effects of the relationship between mattering and depressive symptoms in the postpartum. *Note*. Line values represent non-standard regression coefficients. In the line that links mattering to depressive symptomatology, the value outside the parenthesis represents the total effect of mattering in depressive symptomatology. The value within parentheses represents the direct effect, estimated from the analyses of bootstrapping, of mattering in depressive symptomatology, controlling loneliness. * *p* < 0.001, ** *p* < 0.01.

**Table 1 ijerph-19-11671-t001:** Associations between mattering, loneliness, and depressive symptoms (Pearson correlations).

	Mattering (GMS)	Loneliness (ULS-6)	Depressive Symptoms (EPDS)
**GMS**	-		
**ULS-6**	−0.69 **	-	
**EPDS**	−0.48 **	0.56 **	-

** *p* < 0.001.

**Table 2 ijerph-19-11671-t002:** Associations between some sociodemographic, clinical, and infant’s characteristics (age, educational level, residence, baby’s age, number of children, and gestation week at birth) and depressive symptoms (Spearman correlations).

	Age	Education Level	Residence	Baby’s Age	Number of Children	Gestation Week at Birth	EPDS
Age	-						
Educational level	0.24 **	-					
Residence	−0.08	−0.13 **	-				
Baby’s age	0.08	0.01	−0.08	-			
Number of children	0.35 **	−0.03	−0.02	−0.02	-		
Gestation week at birth	−0.07	0.02	−0.01	0.03	0.03	-	
EPDS	−0.10 *	−0.17 **	−0.01	0.03	−0.09 *	0.00	-

* *p* < 0.05, ** *p* < 0.001.

## Data Availability

The data presented in this study are available on request from the corresponding author.

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
