# Peer review of "Mattering and Depressive Symptoms in Portuguese Postpartum Women: The Indirect Effect of Loneliness"

_ijerph, 2022, doi:10.3390/ijerph191811671_

Round 1
Reviewer 1 Report
This study explored the relationship between matter, loneliness, and postpartum depression, revealing the mediating role of loneliness. The biggest problem with this manuscript is that the model wasn’t basing on a specific theory, but staying at the correlation between the 3 variables to propose a mediating model; and therefore the theoretical implications are not strong. All three key variables in the manuscript are latent variables for multiple projects, and the authors treat them as explicitly observed variables. It is recommended to consider them as latent variables for building the model. In addition, education level and number of children are treated as covariates in this paper, which are not marked in the model diagram (results section). Line 40-45 The content of lines 40 to 45 is not directly related to the research topic and seems cumbersome. Line 79-82 This section should be moved to the discussion section. Line 90 This sentence is confusing
Reviewer 2 Report
Comments to the Authors
Thank you for your description and analysis of your study aimed to understand the relationship between mattering, loneliness, and depressive symptoms among Portuguese mothers. There are not enough studies about mattering/ loneliness and postpartum depression. I think this topic is relevant, in particular after the COVID-19 pandemic. Due to loneliness and mattering are risk factors for perinatal women, it's important to evaluate/ screen these factors during pregnancy and postpartum and provide intervention programs for mothers.
The manuscript is well written. The introduction of adequate and sufficient information even my comments. The methods are well explicated and reproducible. The results are clear but can be improved, I suggest adding tables with all the variables and the analysis. And the discussion derives from the results adequately but with minor comments.
In order to improve the manuscript, I suggest only some minor revisions.
See please my comments and suggestions highlighted in the manuscript.
